# Extending Gossip Algorithms to Distributed Estimation of $U$-Statistics

**Igor Colin, Joseph Salmon, Stéphan Clémençon**
LTCI, CNRS, Télécom ParisTech
Université Paris-Saclay
75013 Paris, France
`first.last@telecom-paristech.fr`

**Aurélien Bellet**
Magnet Team
INRIA Lille - Nord Europe
59650 Villeneuve d'Ascq, France
`aurelien.bellet@inria.fr`

## Abstract

Efficient and robust algorithms for decentralized estimation in networks are essential to many distributed systems. Whereas distributed estimation of sample mean statistics has been the subject of a good deal of attention, computation of $U$-statistics, relying on more expensive averaging over pairs of observations, is a less investigated area. Yet, such data functionals are essential to describe global properties of a statistical population, with important examples including Area Under the Curve, empirical variance, Gini mean difference and within-cluster point scatter. This paper proposes new synchronous and asynchronous randomized gossip algorithms which simultaneously propagate data across the network and maintain local estimates of the $U$-statistic of interest. We establish convergence rate bounds of $O(1/t)$ and $O(\log t/t)$ for the synchronous and asynchronous cases respectively, where $t$ is the number of iterations, with explicit data and network dependent terms. Beyond favorable comparisons in terms of rate analysis, numerical experiments provide empirical evidence the proposed algorithms surpasses the previously introduced approach.

## 1 Introduction

Decentralized computation and estimation have many applications in sensor and peer-to-peer networks as well as for extracting knowledge from massive information graphs such as interlinked Web documents and on-line social media. Algorithms running on such networks must often operate under tight constraints: the nodes forming the network cannot rely on a centralized entity for communication and synchronization, without being aware of the global network topology and/or have limited resources (computational power, memory, energy). Gossip algorithms [19, 18, 5], where each node exchanges information with at most one of its neighbors at a time, have emerged as a simple yet powerful technique for distributed computation in such settings. Given a data observation on each node, gossip algorithms can be used to compute averages or sums of functions of the data that are *separable across observations* (see for example [10, 2, 15, 11, 9] and references therein). Unfortunately, these algorithms cannot be used to efficiently compute quantities that take the form of an average over *pairs of observations*, also known as $U$-statistics [12]. Among classical $U$-statistics used in machine learning and data mining, one can mention, among others: the sample variance, the Area Under the Curve (AUC) of a classifier on distributed data, the Gini mean difference, the Kendall tau rank correlation coefficient, the within-cluster point scatter and several statistical hypothesis test statistics such as Wilcoxon Mann-Whitney [14].

In this paper, we propose randomized synchronous and asynchronous gossip algorithms to efficiently compute a $U$-statistic, in which each node maintains a local estimate of the quantity of interest throughout the execution of the algorithm. Our methods rely on two types of iterative information exchange in the network: propagation of local observations across the network, and averaging of lo-

cal estimates. We show that the local estimates generated by our approach converge in expectation to the value of the $U$-statistic at rates of $O(1/t)$ and $O(\log t/t)$ for the synchronous and asynchronous versions respectively, where $t$ is the number of iterations. These convergence bounds feature data-dependent terms that reflect the hardness of the estimation problem, and network-dependent terms related to the spectral gap of the network graph [3], showing that our algorithms are faster on well-connected networks. The proofs rely on an original reformulation of the problem using "phantom nodes", *i.e.,* on additional nodes that account for data propagation in the network. Our results largely improve upon those presented in [17]: in particular, we achieve faster convergence together with lower memory and communication costs. Experiments conducted on AUC and within-cluster point scatter estimation using real data confirm the superiority of our approach.

The rest of this paper is organized as follows. Section 2 introduces the problem of interest as well as relevant notation. Section 3 provides a brief review of the related work in gossip algorithms. We then describe our approach along with the convergence analysis in Section 4, both in the synchronous and asynchronous settings. Section 5 presents our numerical results.

## 2 Background

### 2.1 Definitions and Notations

For any integer $p > 0$, we denote by $[p]$ the set $\{1, \ldots, p\}$ and by $|F|$ the cardinality of any finite set $F$. We represent a network of size $n > 0$ as an undirected graph $G = (V, E)$, where $V = [n]$ is the set of vertices and $E \subseteq V \times V$ the set of edges. We denote by $A(G)$ the adjacency matrix related to the graph $G$, that is for all $(i, j) \in V^2$, $[A(G)]_{ij} = 1$ if and only if $(i, j) \in E$. For any node $i \in V$, we denote its degree by $d_i = |\{j : (i, j) \in E\}|$. We denote by $L(G)$ the graph Laplacian of $G$, defined by $L(G) = D(G) - A(G)$ where $D(G) = \text{diag}(d_1, \ldots, d_n)$ is the matrix of degrees. A graph $G = (V, E)$ is said to be connected if for all $(i, j) \in V^2$ there exists a path connecting $i$ and $j$; it is bipartite if there exist $S, T \subset V$ such that $S \cup T = V$, $S \cap T = \emptyset$ and $E \subseteq (S \times T) \cup (T \times S)$.

A matrix $M \in \mathbb{R}^{n \times n}$ is nonnegative (resp. positive) if and only if for all $(i, j) \in [n]^2$, $[M]_{ij} \geq 0$, (resp. $[M]_{ij} > 0$). We write $M \geq 0$ (resp. $M > 0$) when this holds. The transpose of $M$ is denoted by $M^\top$. A matrix $P \in \mathbb{R}^{n \times n}$ is stochastic if and only if $P \geq 0$ and $P\mathbf{1}_n = \mathbf{1}_n$, where $\mathbf{1}_n = (1, \ldots, 1)^\top \in \mathbb{R}^n$. The matrix $P \in \mathbb{R}^{n \times n}$ is bi-stochastic if and only if $P$ and $P^\top$ are stochastic. We denote by $I_n$ the identity matrix in $\mathbb{R}^{n \times n}$, $(e_1, \ldots, e_n)$ the standard basis in $\mathbb{R}^n$, $\mathbb{I}_{\{\mathcal{E}\}}$ the indicator function of an event $\mathcal{E}$ and $\| \cdot \|$ the usual $\ell_2$ norm.

### 2.2 Problem Statement

Let $\mathcal{X}$ be an input space and $(X_1, \ldots, X_n) \in \mathcal{X}^n$ a sample of $n \geq 2$ points in that space. We assume $\mathcal{X} \subseteq \mathbb{R}^d$ for some $d > 0$ throughout the paper, but our results straightforwardly extend to the more general setting. We denote as $\mathbf{X} = (X_1, \ldots, X_n)^\top$ the design matrix. Let $H : \mathcal{X} \times \mathcal{X} \to \mathbb{R}$ be a measurable function, symmetric in its two arguments and with $H(X, X) = 0$, $\forall X \in \mathcal{X}$. We consider the problem of estimating the following quantity, known as a degree two $U$-statistic [12]:[1]

$$\hat{U}_n(H) = \frac{1}{n^2} \sum_{i,j=1}^n H(X_i, X_j). \tag{1}$$

In this paper, we illustrate the interest of $U$-statistics on two applications, among many others. The first one is the within-cluster point scatter [4], which measures the clustering quality of a partition $\mathcal{P}$ of $\mathcal{X}$ as the average distance between points in each cell $\mathcal{C} \in \mathcal{P}$. It is of the form (1) with

$$H_{\mathcal{P}}(X, X') = \|X - X'\| \cdot \sum_{\mathcal{C} \in \mathcal{P}} \mathbb{I}_{\{(X, X') \in \mathcal{C}^2\}}. \tag{2}$$

We also study the AUC measure [8]. For a given sample $(X_1, \ell_1), \ldots, (X_n, \ell_n)$ on $\mathcal{X} \times \{-1, +1\}$, the AUC measure of a linear classifier $\theta \in \mathbb{R}^{d-1}$ is given by:

$$\text{AUC}(\theta) = \frac{\sum_{1 \leq i,j \leq n} (1 - \ell_i \ell_j) \mathbb{I}_{\{\ell_i(\theta^\top X_i) > -\ell_j(\theta^\top X_j)\}}}{4 \left( \sum_{1 \leq i \leq n} \mathbb{I}_{\{\ell_i=1\}} \right) \left( \sum_{1 \leq i \leq n} \mathbb{I}_{\{\ell_i=-1\}} \right)}. \tag{3}$$

**Algorithm 1** GoSta-sync: a synchronous gossip algorithm for computing a $U$-statistic

---

**Require:** Each node $k$ holds observation $X_k$
1: Each node $k$ initializes its auxiliary observation $Y_k = X_k$ and its estimate $Z_k = 0$
2: **for** $t = 1, 2, \ldots$ **do**
3:    **for** $p = 1, \ldots, n$ **do**
4:       Set $Z_p \leftarrow \frac{t-1}{t} Z_p + \frac{1}{t} H(X_p, Y_p)$
5:    **end for**
6:    Draw $(i, j)$ uniformly at random from $E$
7:    Set $Z_i, Z_j \leftarrow \frac{1}{2}(Z_i + Z_j)$
8:    Swap auxiliary observations of nodes $i$ and $j$: $Y_i \leftrightarrow Y_j$
9: **end for**

---

This score is the probability for a classifier to rank a positive observation higher than a negative one.

We focus here on the *decentralized setting*, where the data sample is partitioned across a set of nodes in a network. For simplicity, we assume $V = [n]$ and each node $i \in V$ only has access to a single data observation $X_i$.[2] We are interested in estimating (1) efficiently using a gossip algorithm.

## 3 Related Work

Gossip algorithms have been extensively studied in the context of decentralized averaging in networks, where the goal is to compute the average of $n$ real numbers ($\mathcal{X} = \mathbb{R}$):

$$\bar{X}_n = \frac{1}{n} \sum_{i=1}^{n} X_i = \frac{1}{n} \mathbf{X}^\top \mathbf{1}_n. \tag{4}$$

One of the earliest work on this canonical problem is due to [19], but more efficient algorithms have recently been proposed, see for instance [10, 2]. Of particular interest to us is the work of [2], which introduces a randomized gossip algorithm for computing the empirical mean (4) in a context where nodes wake up asynchronously and simply average their local estimate with that of a randomly chosen neighbor. The communication probabilities are given by a stochastic matrix $P$, where $p_{ij}$ is the probability that a node $i$ selects neighbor $j$ at a given iteration. As long as the network graph is connected and non-bipartite, the local estimates converge to (4) at a rate $O(e^{-ct})$ where the constant $c$ can be tied to the spectral gap of the network graph [3], showing faster convergence for well-connected networks.[3] Such algorithms can be extended to compute other functions such as maxima and minima, or sums of the form $\sum_{i=1}^{n} f(X_i)$ for some function $f : \mathcal{X} \to \mathbb{R}$ (as done for instance in [15]). Some work has also gone into developing faster gossip algorithms for poorly connected networks, assuming that nodes know their (partial) geographic location [6, 13]. For a detailed account of the literature on gossip algorithms, we refer the reader to [18, 5].

However, existing gossip algorithms cannot be used to efficiently compute (1) as it depends on *pairs* of observations. To the best of our knowledge, this problem has only been investigated in [17]. Their algorithm, coined U2-gossip, achieves $O(1/t)$ convergence rate but has several drawbacks. First, each node must store two auxiliary observations, and two pairs of nodes must exchange an observation at each iteration. For high-dimensional problems (large $d$), this leads to a significant memory and communication load. Second, the algorithm is not asynchronous as every node must update its estimate at each iteration. Consequently, nodes must have access to a global clock, which is often unrealistic in practice. In the next section, we introduce new synchronous and asynchronous algorithms with faster convergence as well as smaller memory and communication cost per iteration.

## 4 GoSta Algorithms

In this section, we introduce gossip algorithms for computing (1). Our approach is based on the observation that $\hat{U}_n(H) = 1/n \sum_{i=1}^{n} \bar{h}_i$, with $\bar{h}_i = 1/n \sum_{j=1}^{n} H(X_i, X_j)$, and we write $\bar{\mathbf{h}} = (\bar{h}_1, \ldots, \bar{h}_n)^\top$. The goal is thus similar to the usual distributed averaging problem (4), with the

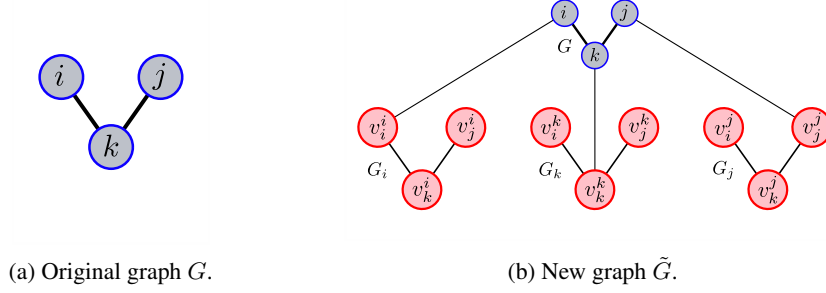

(a) Original graph $G$.  (b) New graph $\tilde{G}$.

Figure 1: Comparison of original network and "phantom network".

key difference that each local value $\overline{h}_i$ is itself an average depending on the entire data sample. Consequently, our algorithms will combine two steps at each iteration: a data propagation step to allow each node $i$ to estimate $\overline{h}_i$, and an averaging step to ensure convergence to the desired value $\hat{U}_n(H)$. We first present the algorithm and its analysis for the (simpler) synchronous setting in Section 4.1, before introducing an asynchronous version (Section 4.2).

## 4.1 Synchronous Setting

In the synchronous setting, we assume that the nodes have access to a global clock so that they can all update their estimate at each time instance. We stress that the nodes need not to be aware of the global network topology as they will only interact with their direct neighbors in the graph.

Let us denote by $Z_k(t)$ the (local) estimate of $\hat{U}_n(H)$ by node $k$ at iteration $t$. In order to propagate data across the network, each node $k$ maintains an auxiliary observation $Y_k$, initialized to $X_k$. Our algorithm, coined GoSta, goes as follows. At each iteration, each node $k$ updates its local estimate by taking the running average of $Z_k(t)$ and $H(X_k, Y_k)$. Then, an edge of the network is drawn uniformly at random, and the corresponding pair of nodes average their local estimates and swap their auxiliary observations. The observations are thus each performing a random walk (albeit coupled) on the network graph. The full procedure is described in Algorithm 1.

In order to prove the convergence of Algorithm 1, we consider an equivalent reformulation of the problem which allows us to model the data propagation and the averaging steps separately. Specifically, for each $k \in V$, we define a phantom $G_k = (V_k, E_k)$ of the original network $G$, with $V_k = \{v_i^k; 1 \leq i \leq n\}$ and $E_k = \{(v_i^k, v_j^k); (i, j) \in E\}$. We then create a new graph $\tilde{G} = (\tilde{V}, \tilde{E})$ where each node $k \in V$ is connected to its counterpart $v_k^k \in V_k$:

$$\begin{cases} \tilde{V} & = & V \cup (\cup_{k=1}^n V_k) \\ \tilde{E} & = & E \cup (\cup_{k=1}^n E_k) \cup \{(k, v_k^k); k \in V\} \end{cases}$$

The construction of $\tilde{G}$ is illustrated in Figure 1. In this new graph, the nodes $V$ from the original network will hold the estimates $Z_1(t), \ldots, Z_n(t)$ as described above. The role of each $G_k$ is to simulate the data propagation in the original graph $G$. For $i \in [n]$, $v_i^k \in V^k$ initially holds the value $H(X_k, X_i)$. At each iteration, we draw a random edge $(i, j)$ of $G$ and nodes $v_i^k$ and $v_j^k$ swap their value for all $k \in [n]$. To update its estimate, each node $k$ will use the current value at $v_k^k$.

We can now represent the system state at iteration $t$ by a vector $\mathbf{S}(t) = (\mathbf{S}_1(t)^\top, \mathbf{S}_2(t)^\top)^\top \in \mathbb{R}^{n+n^2}$. The first $n$ coefficients, $\mathbf{S}_1(t)$, are associated with nodes in $V$ and correspond to the estimate vector $\mathbf{Z}(t) = [Z_1(t), \ldots, Z_n(t)]^\top$. The last $n^2$ coefficients, $\mathbf{S}_2(t)$, are associated with nodes in $(V_k)_{1 \leq k \leq n}$ and represent the data propagation in the network. Their initial value is set to $\mathbf{S}_2(0) = (e_1^\top \mathbf{H}, \ldots, e_n^\top \mathbf{H})$ so that for any $(k, l) \in [n]^2$, node $v_l^k$ initially stores the value $H(X_k, X_l)$.

**Remark 1.** The "phantom network" $\tilde{G}$ is of size $O(n^2)$, but we stress the fact that it is used solely as a tool for the convergence analysis: Algorithm 1 operates on the original graph $G$.

The transition matrix of this system accounts for three events: the *averaging step* (the action of $G$ on itself), the *data propagation* (the action of $G_k$ on itself for all $k \in V$) and the *estimate update*

(the action of $G_k$ on node $k$ for all $k \in V$). At a given step $t > 0$, we are interested in characterizing the transition matrix $M(t)$ such that $\mathbb{E}[\mathbf{S}(t+1)] = M(t)\mathbb{E}[\mathbf{S}(t)]$. For the sake of clarity, we write $M(t)$ as an upper block-triangular $(n + n^2) \times (n + n^2)$ matrix:

$$M(t) = \begin{pmatrix} M_1(t) & M_2(t) \\ 0 & M_3(t) \end{pmatrix}, \tag{5}$$

with $M_1(t) \in \mathbb{R}^{n \times n}$, $M_2(t) \in \mathbb{R}^{n \times n^2}$ and $M_3(t) \in \mathbb{R}^{n^2 \times n^2}$. The bottom left part is necessarily 0, because $G$ does not influence any $G_k$. The upper left $M_1(t)$ block corresponds to the averaging step; therefore, for any $t > 0$, we have:

$$M_1(t) = \frac{t-1}{t} \cdot \frac{1}{|E|} \sum_{(i,j) \in E} \left( I_n - \frac{1}{2}(e_i - e_j)(e_i - e_j)^\top \right) = \frac{t-1}{t} W_2(G),$$

where for any $\alpha > 1$, $W_\alpha(G)$ is defined by:

$$W_\alpha(G) = \frac{1}{|E|} \sum_{(i,j) \in E} \left( I_n - \frac{1}{\alpha}(e_i - e_j)(e_i - e_j)^\top \right) = I_n - \frac{2}{\alpha|E|} L(G). \tag{6}$$

Furthermore, $M_2(t)$ and $M_3(t)$ are defined as follows:

$$M_2(t) = \frac{1}{t} \underbrace{\begin{pmatrix} e_1^\top & 0 & \cdots & 0 \\ 0 & \ddots & & \vdots \\ \vdots & & \ddots & 0 \\ 0 & \cdots & 0 & e_n^\top \end{pmatrix}}_{B} \quad \text{and} \quad M_3(t) = \underbrace{\begin{pmatrix} W_1(G) & 0 & \cdots & 0 \\ 0 & \ddots & & \vdots \\ \vdots & & \ddots & \vdots \\ 0 & \cdots & 0 & W_1(G) \end{pmatrix}}_{C},$$

where $M_2(t)$ is a block diagonal matrix corresponding to the observations being propagated, and $M_3(t)$ represents the estimate update for each node $k$. Note that $M_3(t) = W_1(G) \otimes I_n$ where $\otimes$ is the Kronecker product.

We can now describe the expected state evolution. At iteration $t = 0$, one has:

$$\mathbb{E}[S(1)] = M(1)\mathbb{E}[S(0)] = M(1)S(0) = \begin{pmatrix} 0 & B \\ 0 & C \end{pmatrix} \begin{pmatrix} 0 \\ \mathbf{S}_2(0) \end{pmatrix} = \begin{pmatrix} B\mathbf{S}_2(0) \\ C\mathbf{S}_2(0) \end{pmatrix}. \tag{7}$$

Using recursion, we can write:

$$\mathbb{E}[\mathbf{S}(t)] = M(t)M(t-1)\ldots M(1)\mathbf{S}(0) = \begin{pmatrix} \frac{1}{t} \sum_{s=1}^{t} W_2(G)^{t-s} BC^{s-1}\mathbf{S}_2(0) \\ C^t \mathbf{S}_2(0) \end{pmatrix}. \tag{8}$$

Therefore, in order to prove the convergence of Algorithm 1, one needs to show that $\lim_{t \to +\infty} \frac{1}{t} \sum_{s=1}^{t} W_2(G)^{t-s} BC^{s-1}\mathbf{S}_2(0) = \hat{U}_n(H)\mathbf{1}_n$. We state this precisely in the next theorem.

**Theorem 1.** *Let $G$ be a connected and non-bipartite graph with $n$ nodes, $\mathbf{X} \in \mathbb{R}^{n \times d}$ a design matrix and $(\mathbf{Z}(t))$ the sequence of estimates generated by Algorithm 1. For all $k \in [n]$, we have:*

$$\lim_{t \to +\infty} \mathbb{E}[Z_k(t)] = \frac{1}{n^2} \sum_{1 \leq i,j \leq n} H(X_i, X_j) = \hat{U}_n(H). \tag{9}$$

*Moreover, for any $t > 0$,*

$$\left\| \mathbb{E}[\mathbf{Z}(t)] - \hat{U}_n(H)\mathbf{1}_n \right\| \leq \frac{1}{ct} \left\| \overline{\mathbf{h}} - \hat{U}_n(H)\mathbf{1}_n \right\| + \left( \frac{2}{ct} + e^{-ct} \right) \left\| \mathbf{H} - \overline{\mathbf{h}}\mathbf{1}_n^\top \right\|,$$

*where $c = c(G) := 1 - \lambda_2(2)$ and $\lambda_2(2)$ is the second largest eigenvalue of $W_2(G)$.*

*Proof.* See supplementary material. □

Theorem 1 shows that the local estimates generated by Algorithm 1 converge to $\hat{U}_n(H)$ at a rate $O(1/t)$. Furthermore, the constants reveal the rate dependency on the particular problem instance. Indeed, the two norm terms are *data-dependent* and quantify the difficulty of the estimation problem itself through a dispersion measure. In contrast, $c(G)$ is a *network-dependent* term since $1 - \lambda_2(2) = \beta_{n-1}/|E|$, where $\beta_{n-1}$ is the second smallest eigenvalue of the graph Laplacian $L(G)$ (see Lemma 1 in the supplementary material). The value $\beta_{n-1}$ is also known as the spectral gap of $G$ and graphs with a larger spectral gap typically have better connectivity [3]. This will be illustrated in Section 5.

---

**Algorithm 2** GoSta-async: an asynchronous gossip algorithm for computing a $U$-statistic

---
**Require:** Each node $k$ holds observation $X_k$ and $p_k = 2d_k/|E|$
1: Each node $k$ initializes $Y_k = X_k$, $Z_k = 0$ and $m_k = 0$
2: **for** $t = 1, 2, \ldots$ **do**
3:     Draw $(i,j)$ uniformly at random from $E$
4:     Set $m_i \leftarrow m_i + 1/p_i$ and $m_j \leftarrow m_j + 1/p_j$
5:     Set $Z_i, Z_j \leftarrow \frac{1}{2}(Z_i + Z_j)$
6:     Set $Z_i \leftarrow (1 - \frac{1}{p_i m_i})Z_i + \frac{1}{p_i m_i}H(X_i, Y_i)$
7:     Set $Z_j \leftarrow (1 - \frac{1}{p_j m_j})Z_j + \frac{1}{p_j m_j}H(X_j, Y_j)$
8:     Swap auxiliary observations of nodes $i$ and $j$: $Y_i \leftrightarrow Y_j$
9: **end for**

---

**Comparison to U2-gossip.** To estimate $\hat{U}_n(H)$, U2-gossip [17] does not use averaging. Instead, each node $k$ requires two auxiliary observations $Y_k^{(1)}$ and $Y_k^{(2)}$ which are both initialized to $X_k$. At each iteration, each node $k$ updates its local estimate by taking the running average of $Z_k$ and $H(Y_k^{(1)}, Y_k^{(2)})$. Then, two random edges are selected: the nodes connected by the first (resp. second) edge swap their first (resp. second) auxiliary observations. A precise statement of the algorithm is provided in the supplementary material. U2-gossip has several drawbacks compared to GoSta: it requires initiating communication between two pairs of nodes at each iteration, and the amount of communication and memory required is higher (especially when data is high-dimensional). Furthermore, applying our convergence analysis to U2-gossip, we obtain the following refined rate:[4]

$$\left\| \mathbb{E}[\mathbf{Z}(t)] - \hat{U}_n(H)\mathbf{1}_n \right\| \leq \frac{\sqrt{n}}{t}\left( \frac{2}{1 - \lambda_2(1)}\left\| \overline{\mathbf{h}} - \hat{U}_n(H)\mathbf{1}_n \right\| + \frac{1}{1 - \lambda_2(1)^2}\left\| \mathbf{H} - \overline{\mathbf{h}}\mathbf{1}_n^\top \right\| \right),$$
(10)

where $1 - \lambda_2(1) = 2(1 - \lambda_2(2)) = 2c(G)$ and $\lambda_2(1)$ is the second largest eigenvalue of $W_1(G)$. The advantage of propagating two observations in U2-gossip is seen in the $1/(1 - \lambda_2(1)^2)$ term, however the absence of averaging leads to an overall $\sqrt{n}$ factor. Intuitively, this is because nodes do not benefit from each other's estimates. In practice, $\lambda_2(2)$ and $\lambda_2(1)$ are close to 1 for reasonably-sized networks (for instance, $\lambda_2(2) = 1 - 1/n$ for the complete graph), so the square term does not provide much gain and the $\sqrt{n}$ factor dominates in (10). We thus expect U2-gossip to converge slower than GoSta, which is confirmed by the numerical results presented in Section 5.

## 4.2 Asynchronous Setting

In practical settings, nodes may not have access to a global clock to synchronize the updates. In this section, we remove the global clock assumption and propose a fully asynchronous algorithm where each node has a local clock, ticking at a rate 1 Poisson process. Yet, local clocks are i.i.d. so one can use an equivalent model with a global clock ticking at a rate $n$ Poisson process and a random edge draw at each iteration, as in synchronous setting (one may refer to [2] for more details on clock modeling). However, at a given iteration, the estimate update step now only involves the selected pair of nodes. Therefore, the nodes need to maintain an estimate of the current iteration number to ensure convergence to an unbiased estimate of $\hat{U}_n(H)$. Hence for all $k \in [n]$, let $p_k \in [0,1]$ denote the probability of node $k$ being picked at any iteration. With our assumption that nodes activate with a uniform distribution over $E$, $p_k = 2d_k/|E|$. Moreover, the number of times a node $k$ has been selected at a given iteration $t > 0$ follows a binomial distribution with parameters $t$ and $p_k$. Let us define $m_k(t)$ such that $m_k(0) = 0$ and for $t > 0$:

$$m_k(t) = \begin{cases} m_k(t-1) + \frac{1}{p_k} & \text{if } k \text{ is picked at iteration t,} \\ m_k(t-1) & \text{otherwise.} \end{cases}$$
(11)

For any $k \in [n]$ and any $t > 0$, one has $\mathbb{E}[m_k(t)] = t \times p_k \times 1/p_k = t$. Therefore, given that every node knows its degree and the total number of edges in the network, the iteration estimates are unbiased. We can now give an asynchronous version of GoSta, as stated in Algorithm 2.

To show that local estimates converge to $\hat{U}_n(H)$, we use a similar model as in the synchronous setting. The time dependency of the transition matrix is more complex ; so is the upper bound.

| Dataset | Complete graph | Watts-Strogatz | 2d-grid graph |
|---|---|---|---|
| Wine Quality ($n = 1599$) | $6.26 \cdot 10^{-4}$ | $2.72 \cdot 10^{-5}$ | $3.66 \cdot 10^{-6}$ |
| SVMguide3 ($n = 1260$) | $7.94 \cdot 10^{-4}$ | $5.49 \cdot 10^{-5}$ | $6.03 \cdot 10^{-6}$ |

Table 1: Value of $1 - \lambda_2(2)$ for each network.

**Theorem 2.** *Let $G$ be a connected and non bipartite graph with $n$ nodes, $\mathbf{X} \in \mathbb{R}^{n \times d}$ a design matrix and $(\mathbf{Z}(t))$ the sequence of estimates generated by Algorithm 2. For all $k \in [n]$, we have:*

$$\lim_{t \to +\infty} \mathbb{E}[Z_k(t)] = \frac{1}{n^2} \sum_{1 \leq i,j \leq n} H(X_i, X_j) = \hat{U}_n(H). \tag{12}$$

*Moreover, there exists a constant $c'(G) > 0$ such that, for any $t > 1$,*

$$\left\| \mathbb{E}[\mathbf{Z}(t)] - \hat{U}_n(H)\mathbf{1}_n \right\| \leq c'(G) \cdot \frac{\log t}{t} \|\mathbf{H}\|. \tag{13}$$

*Proof.* See supplementary material. □

**Remark 2.** Our methods can be extended to the situation where nodes contain multiple observations: when drawn, a node will pick a random auxiliary observation to swap. Similar convergence results are achieved by splitting each node into a set of nodes, each containing only one observation and new edges weighted judiciously.

## 5 Experiments

In this section, we present two applications on real datasets: the decentralized estimation of the Area Under the ROC Curve (AUC) and of the within-cluster point scatter. We compare the performance of our algorithms to that of U2-gossip [17] — see supplementary material for additional comparisons to some baseline methods. We perform our simulations on the three types of network described below (corresponding values of $1 - \lambda_2(2)$ are shown in Table 1).

• *Complete graph:* This is the case where all nodes are connected to each other. It is the ideal situation in our framework, since any pair of nodes can communicate directly. For a complete graph $G$ of size $n > 0$, $1 - \lambda_2(2) = 1/n$, see [1, Ch.9] or [3, Ch.1] for details.

• *Two-dimensional grid:* Here, nodes are located on a 2D grid, and each node is connected to its four neighbors on the grid. This network offers a regular graph with isotropic communication, but its diameter ($\sqrt{n}$) is quite high, especially in comparison to usual scale-free networks.

• *Watts-Strogatz:* This random network generation technique is introduced in [20] and allows us to create networks with various communication properties. It relies on two parameters: the average degree of the network $k$ and a rewiring probability $p$. In expectation, the higher the rewiring probability, the better the connectivity of the network. Here, we use $k = 5$ and $p = 0.3$ to achieve a connectivity compromise between the complete graph and the two-dimensional grid.

**AUC measure.** We first focus on the AUC measure of a linear classifier $\theta$ as defined in (3). We use the SMVguide3 binary classification dataset which contains $n = 1260$ points in $d = 23$ dimensions.[5] We set $\theta$ to the difference between the class means. For each generated network, we perform 50 runs of GoSta-sync (Algorithm 1) and U2-gossip. The top row of Figure 2 shows the evolution over time of the average relative error and the associated standard deviation *across nodes* for both algorithms on each type of network. On average, GoSta-sync outperforms U2-gossip on every network. The variance of the estimates across nodes is also lower due to the averaging step. Interestingly, the performance gap between the two algorithms is greatly increasing early on, presumably because the exponential term in the convergence bound of GoSta-sync is significant in the first steps.

**Within-cluster point scatter.** We then turn to the within-cluster point scatter defined in (2). We use the Wine Quality dataset which contains $n = 1599$ points in $d = 12$ dimensions, with a total of $K = 11$ classes.[6] We focus on the partition $\mathcal{P}$ associated to class centroids and run the aforementioned

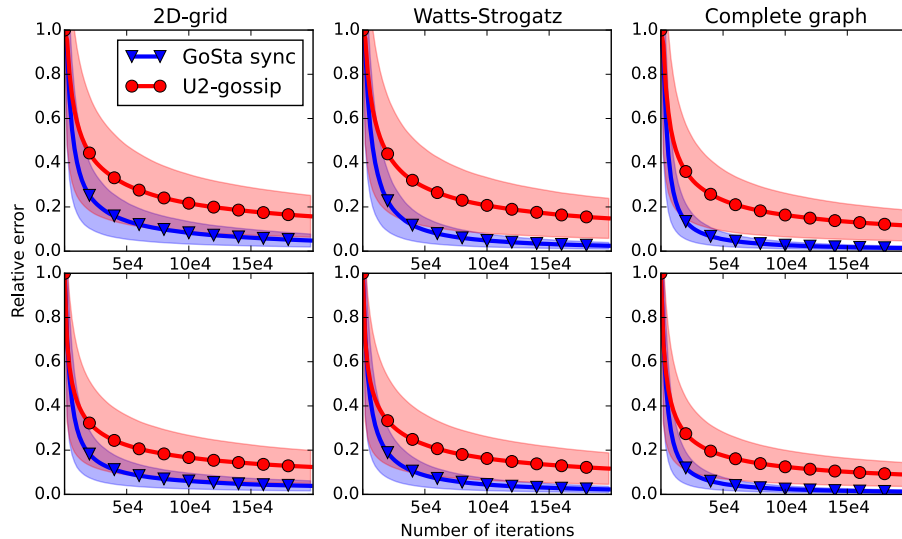

Figure 2: Evolution of the average relative error (solid line) and its standard deviation (filled area) with the number of iterations for U2-gossip (red) and Algorithm 1 (blue) on the SVMguide3 dataset (top row) and the Wine Quality dataset (bottom row).

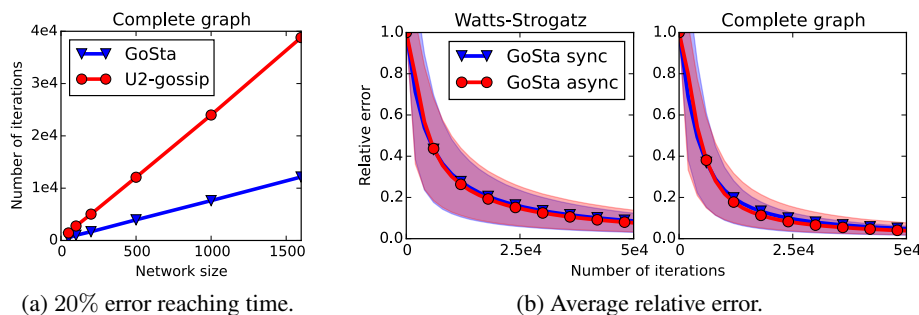

(a) 20% error reaching time.                (b) Average relative error.

Figure 3: Panel (a) shows the average number of iterations needed to reach an relative error below $0.2$, for several network sizes $n \in [50, 1599]$. Panel (b) compares the relative error (solid line) and its standard deviation (filled area) of synchronous (blue) and asynchronous (red) versions of GoSta.

methods 50 times. The results are shown in the bottom row of Figure 2. As in the case of AUC, GoSta-sync achieves better perfomance on all types of networks, both in terms of average error and variance. In Figure 3a, we show the average time needed to reach a $0.2$ relative error on a complete graph ranging from $n = 50$ to $n = 1599$. As predicted by our analysis, the performance gap widens in favor of GoSta as the size of the graph increases. Finally, we compare the performance of GoSta-sync and GoSta-async (Algorithm 2) in Figure 3b. Despite the slightly worse theoretical convergence rate for GoSta-async, both algorithms have comparable performance in practice.

# 6  Conclusion

We have introduced new synchronous and asynchronous randomized gossip algorithms to compute statistics that depend on pairs of observations ($U$-statistics). We have proved the convergence rate in both settings, and numerical experiments confirm the practical interest of the proposed algorithms. In future work, we plan to investigate whether adaptive communication schemes (such as those of [6, 13]) can be used to speed-up our algorithms. Our contribution could also be used as a building block for decentralized *optimization* of $U$-statistics, extending for instance the approaches of [7, 16].

**Acknowledgements**   This work was supported by the chair Machine Learning for Big Data of Télécom ParisTech, and was conducted when A. Bellet was affiliated with Télécom ParisTech.

## Footnotes

[1]We point out that the usual definition of $U$-statistic differs slightly from (1) by a factor of $n/(n-1)$.

[2] Our results generalize to the case where each node holds a subset of the observations (see Section 4).

[3] For the sake of completeness, we provide an analysis of this algorithm in the supplementary material.

[4]The proof can be found in the supplementary material.

[5]This dataset is available at `http://mldata.org/repository/data/viewslug/svmguide3/`

[6]This dataset is available at `https://archive.ics.uci.edu/ml/datasets/Wine`

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
