[Supplementary Material]

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

# Appendix

## A Preliminary Results

Here, we state preliminary results on the matrices $W_\alpha(G)$ that will be useful for deriving convergence proofs and compare the algorithms.

First, we characterize the eigenvalues of $W_\alpha(G)$ in terms of those of the graph Laplacian.

**Lemma 1.** Let $G = (V, E)$ be an undirected graph and let $(\beta_i)_{1 \leq i \leq n}$ be the eigenvalues of $L(G)$, sorted in decreasing order. For any $\alpha \geq 1$, we denote as $(\lambda_i(\alpha))_{1 \leq i \leq n}$ the eigenvalues of $W_\alpha(G)$, sorted in decreasing order. Then, for any $1 \leq i \leq n$,

$$\lambda_i(\alpha) = 1 - \frac{2\beta_{n-i+1}}{\alpha|E|}. \tag{14}$$

*Proof.* Let $\alpha \geq 1$. The matrix $W_\alpha(G)$ can be rewritten as follow:

$$W_\alpha(G) = \frac{1}{|E|} \sum_{(i,j) \in E} \left( I_n - \frac{1}{\alpha}(e_i - e_j)(e_i - e_j)^\top \right) \tag{15}$$

$$= I_n - \frac{1}{\alpha|E|} \sum_{(i,j) \in E} (e_i - e_j)(e_i - e_j)^\top = I_n - \frac{2}{\alpha|E|}L(G). \tag{16}$$

Let $\phi_i \in \mathbb{R}^n$ be an eigenvector of $L(G)$ corresponding to an eigenvalue $\beta_i$, then we have:

$$W_\alpha(G)\phi_i = \left( I_n - \frac{2}{\alpha|E|}L(G) \right) \phi_i = \left( 1 - \frac{2}{\alpha|E|}\beta_i \right) \phi_i.$$

Thus, $\phi_i$ is also an eigenvector of $W_\alpha(G)$ for the eigenvalue $1 - \frac{2}{\alpha|E|}\beta_i$ and the result holds. $\square$

The following lemmata provide essential properties on $W_\alpha(G)$ eigenvalues.

**Lemma 2.** Let $n > 0$ and let $G = ([n], E)$ be an undirected graph. If $G$ is connected and non-bipartite, then for any $\alpha \geq 1$, $W_\alpha(G)$ is primitive, *i.e.,* there exists $k > 0$ such that $W_\alpha(G)^k > 0$.

*Proof.* Let $\alpha \geq 1$. For every $(i, j) \in E$, $I_n - \frac{1}{\alpha}(e_i - e_j)(e_i - e_j)^\top$ is nonnegative. Therefore $W_\alpha(G)$ is also nonnegative. For any $1 \leq k < l \leq n$, by definition of $W_\alpha(G)$, one has the following equivalence:

$$([A(G)]_{kl} > 0) \Leftrightarrow ([W_\alpha(G)]_{kl} > 0).$$

By hypothesis, $G$ is connected. Therefore, for any pair of nodes $(k, l) \in V^2$ there exists an integer $s_{kl} > 0$ such that $[A(G)^{s_{kl}}]_{kl} > 0$ so $W_\alpha(G)$ is irreducible. Also, $G$ is non bipartite so similar reasoning can be used to show that $W_\alpha(G)$ is aperiodic.

By the Lattice Theorem (see [3, Th. 4.3, p.75]), for any $1 \leq k, l \leq n$ there exists an integer $m_{kl}$ such that, for any $m \geq m_{kl}$:

$$[W_\alpha(G)^m]_{kl} > 0.$$

Finally, we can define $\bar{m} = \sup_{k,l} m_{kl}$ and observe that $W_\alpha(G)^{\bar{m}} > 0$. $\square$

**Lemma 3.** Let $G = (V, E)$ be a connected and non bipartite graph. Then for any $\alpha \geq 1$,

$$1 = \lambda_1(\alpha) > \lambda_2(\alpha),$$

where $\lambda_1(\alpha)$ and $\lambda_2(\alpha)$ are respectively the largest and the second largest eigenvalue of $W_\alpha(G)$.

*Proof.* Let $\alpha \geq 1$. The matrix $W_\alpha(G)$ is bistochastic, so $\lambda_1(\alpha) = 1$. By Lemma 2, $W_\alpha(G)$ is primitive. Therefore, by the Perron-Frobenius Theorem (see [3, Th. 1.1, p.197]), we can conclude that $\lambda_1(\alpha) > \lambda_2(\alpha)$. $\square$

**Algorithm 3** Gossip algorithm proposed in [2] to compute the standard mean (4)

---

**Require:** Each node $i$ holds observation $X_i$
1: Each node initializes its estimator $Z_i \leftarrow X_i$
2: **for** $t = 1, 2, \ldots$ **do**
3:  Draw $(i, j)$ uniformly at random from $E$
4:  Set $Z_i, Z_j \leftarrow \frac{Z_i + Z_j}{2}$
5: **end for**

---

## B Gossip Algorithm for Standard Averaging

In this part, we provide a description and analysis of the randomized gossip algorithm proposed in [2] for the standard averaging problem (4). The procedure is shown in Algorithm 3 and goes as follows. For each node $k \in [n]$, an estimator $Z_k(t)$ is initialized to the observation of the node $Z_k(0) = X_k$. At each iteration $t > 0$, an edge $(i, j) \in E$ is picked uniformly at random over $E$. Then, the corresponding nodes average their current estimators, while the others remain unchanged:

$$Z_i(t) = Z_j(t) = \frac{Z_i(t-1) + Z_j(t-1)}{2}. \tag{17}$$

The evolution of estimates can be characterized using transition matrices. If an edge $(i, j) \in E$ is picked at iteration $t > 0$, one can re-formulate (17) as:

$$\mathbf{Z}(t) = \left( I_n - \frac{(e_i - e_j)(e_i - e_j)^\top}{2} \right) \mathbf{Z}(t-1),$$

where $\mathbf{Z}(t) = (Z_1(t), \ldots, Z_n(t)) \in \mathbb{R}^n$ is the (global) vector of mean estimates. The edges being drawn uniformly at random, the expected transition matrix is simply $W_2(G)$ with the notation introduced in (6). Then, for all $t > 0$, the expectation of the global estimate at iteration $t$ is characterized recursively by:

$$\mathbb{E}[\mathbf{Z}(t)] = W_2(G)\mathbb{E}[\mathbf{Z}(t-1)] = W_2(G)^t \mathbb{E}[\mathbf{Z}(0)] = W_2(G)^t \mathbf{X},$$

where $\mathbb{E}$ stands for the expectation with respect to the edge sampling process.

We can now state a convergence result for Algorithm 3, rephrasing slightly the results from [2].

**Theorem 3.** *Let us assume that $G$ is connected and non bipartite. Then, for $\mathbf{Z}(t)$ defined in Algorithm 3, we have that for all $k \in [n]$:*

$$\lim_{t \to +\infty} \mathbb{E}[\mathbf{Z}_k(t)] = \bar{X}_n,$$

*Moreover, for any $t > 0$,*

$$\|\mathbb{E}[\mathbf{Z}(t)] - \bar{X}_n \mathbf{1}_n\| \le e^{-ct}\|\mathbf{X} - \bar{X}_n \mathbf{1}_n\|.$$

*where $c = (1 - \lambda_2(2)) > 0$, with $\lambda_2(2)$ the second largest eigenvalue of $W_2(G)$.*

*Proof.* In order to prove the convergence of the estimates in expectation, one has to prove that $W_2(G)\mathbf{X}$ converges to the objective $\bar{X}_n \mathbf{1}_n$.

Remark that $W_2(G)$ is bi-stochastic. Let us denote as $(\lambda_k(2))_{1 \le k \le n}$ and $(\phi_k)_{1 \le k \le n}$ respectively the eigenvalues (sorted in decreasing order) and the corresponding eigenvectors of $W_2(G)$. Lemma 3 indicates that $1 = \lambda_1(2) > \lambda_2(2)$. Therefore, we only need to prove the second assertion since the first one is a direct consequence. Since $W_2(G)$ is symmetric, we can pick eigenvectors that are orthonormal and select $\phi_1$ such that:

$$\phi_1 = \frac{1}{\sqrt{n}} \mathbf{1}_n.$$

Let us also define $D_2 = \text{diag}(\lambda_1(2), \ldots, \lambda_n(2))$ and $P = [\phi_1, \ldots, \phi_n]$. Thus, we have:

$$W_2(G) = PD_2P^\top.$$

Let us split $D_2$ by defining $Q_2 \in \mathbb{R}^{n \times n}$ and $R_2 \in \mathbb{R}^{n \times n}$ such that:

$$\begin{cases} Q_2 &=& \mathrm{diag}(\lambda_1(2), 0, \ldots, 0) \\ R_2 &=& \mathrm{diag}(0, \lambda_2(2), \ldots, \lambda_n(2)) \end{cases}$$

Then, for any $t > 0$, we can write:

$$\mathbb{E}[\mathbf{Z}(t)] = W_2\,(G)^t\,\mathbf{X} = PD_2^t P^\top \mathbf{X} = P\left(Q_2^t + R_2^t\right)P^\top \mathbf{X} = PQ_2^t P^\top \mathbf{X} + PR_2^t P^\top \mathbf{X}.$$

Reminding $\lambda_1(2) = 1$, the first term can be rewritten:

$$PQ_2^t P^\top \mathbf{X} = \phi_1 \phi_1^\top \mathbf{X} = \frac{1}{n}\left(\mathbf{1}_n^\top \mathbf{X}\right)\mathbf{1}_n,$$

which corresponds to the objective $\bar{X}_n \mathbf{1}_n$. We write $\|\!|\cdot|\!\|$ the operator norm of a matrix. Since $R_2 \mathbf{1}_n = 0$, one has for any $t > 0$,

$$\|\mathbb{E}[\mathbf{Z}(t)] - \bar{X}_n \mathbf{1}_n\| = \|W_2\,(G)^t\,\mathbf{X} - \bar{X}_n \mathbf{1}_n\| = \left\|W_2\,(G)^t\,\mathbf{X} - \frac{1}{n}\left(\mathbf{1}_n^\top \mathbf{X}\right)\mathbf{1}_n\right\|$$

$$= \left\|PR_2^t P^\top \mathbf{X}\right\| \le \|\!|PR_2^t P^\top|\!\|\,\|\mathbf{X} - \bar{X}_n \mathbf{1}_n\|$$

$$\le (\lambda_2(2))^t \|\mathbf{X} - \bar{X}_n \mathbf{1}_n\|,$$

and the result holds since $0 \le \lambda_2(2) < 1$. $\qquad\qquad\square$

## C Convergence Proofs for GoSta

### C.1 Proof of Theorem 1 (Synchronous Setting)

Our main goal is to characterize the behavior of $\mathbf{S}_1(t)$, as it corresponds to the estimates $\mathbf{Z}(t)$. As for the standard gossip averaging, our proof relies on the study of eigenvalues and eigenvectors of the transition matrix $M(t)$.

*Proof.* By definition of $M_1(t)$, we have that

$$M_1(t) = \frac{t-1}{t} W_2\,(G) = \frac{t-1}{t} PD_2 P^\top.$$

Similarly, using the fact that $C = W_1\,(G) \otimes I_n$, we have that

$$C = P_c D_c P_c^\top,$$

where $P_c = P \otimes I_n$, $D = D_1 \otimes I_n$ and $D_1 = \mathrm{diag}(\lambda_1(1), \ldots, \lambda_n(1))$. The expected value of $S_1(t)$ can then be rewritten:

$$\mathbb{E}[S_1(t)] = \frac{1}{t}\sum_{s=1}^{t} W_2\,(G)^{t-s} BC^{s-1}\mathbf{S}_2(0) = \frac{1}{t}P\left(\sum_{s=1}^{t} D_2^{t-s}P^\top BP_c D_c^{s-1}\right)P_c^\top \mathbf{S}_2(0). \quad (18)$$

Our objective is to extract the value $\hat{U}_n(H)$ from the expression (18) by separating $\lambda_2(1)$ and $\lambda_1(1)$ from other eigenvalues. Let $Q_c = Q_1 \otimes I_n$ and $R_c = R_1 \otimes I_n$. We can now write $\mathbb{E}[\mathbf{S}_1(t)] = L_1(t) + L_2(t) + L_3(t) + L_4(t)$, where:

$$\begin{cases} L_1(t) &=& \frac{1}{t}\sum_{s=1}^{t} PQ_2^{t-s}P^\top BP_c Q_c P_c^\top \mathbf{S}_2(0), \\ L_2(t) &=& \frac{1}{t}\sum_{s=1}^{t} PR_2^{t-s}P^\top BP_c Q_c P_c^\top \mathbf{S}_2(0), \\ L_3(t) &=& \frac{1}{t}\sum_{s=1}^{t} PQ_2^{t-s}P^\top BP_c R_c^{s-1}P_c^\top \mathbf{S}_2(0), \\ L_4(t) &=& \frac{1}{t}\sum_{s=1}^{t} PR_2^{t-s}P^\top BP_c R_c^{s-1}P_c^\top \mathbf{S}_2(0). \end{cases}$$

We will now show that for any $t > 0$, $L_1(t)$ is actually $\hat{U}_n(H)$. We have:

$$PQ_2 P^\top = \frac{1}{n}\mathbf{1}_n \mathbf{1}_n^\top.$$

Similarly, we have:

$$P_c Q_c P_c^\top = (PQ_1 P^\top) \otimes I_n = \frac{1}{n}(\mathbf{1}_n \mathbf{1}_n^\top) \otimes I_n.$$

Finally, we can write:

$$L_1(t) = PQ_2P^\top BP_cQ_cP_c^\top \mathbf{S}_2(0) = \frac{1}{n^2}\mathbf{1}_n\mathbf{1}_n^\top B\left((\mathbf{1}_n\mathbf{1}_n^\top)\otimes I_n\right)\mathbf{S}_2(0)$$

$$= \frac{1}{n^2}\mathbf{1}_n\mathbf{1}_n^\top(\mathbf{1}_n^\top \otimes I_n)\mathbf{S}_2(0) = \frac{1}{n^2}\mathbf{1}_n\mathbf{1}_{n^2}^\top\mathbf{S}_2(0) = \hat{U}_n(H)\mathbf{1}_n.$$

Let us now focus on the other terms. For $t > 0$, we have:

$$\|L_2(t)\| \leq \frac{1}{t}\sum_{s=1}^{t}\left\|PR_2^{t-s}P^\top BP_cQ_cP_c^\top \mathbf{S}_2(0)\right\|$$

$$= \frac{1}{t}\sum_{s=1}^{t}\left\|PR_2^{t-s}P^\top B\left(\frac{1}{n}\left(\mathbf{1}_n\mathbf{1}_n^\top\right)\otimes I_n\right)\mathbf{S}_2(0)\right\|$$

$$= \frac{1}{t}\sum_{s=1}^{t}\left\|PR_2^{t-s}P^\top\left(\frac{1}{n}\mathbf{1}_n^\top \otimes I_n\right)\mathbf{S}_2(0)\right\|.$$

One has:

$$\left\|\left(\frac{1}{n}\mathbf{1}_n^\top \otimes I_n\right)\mathbf{S}_2(0)\right\|^2 = \sum_{i=1}^{n}\left(\frac{1}{n}\mathbf{1}_n^\top\mathbf{H}e_i\right)^2 = \left\|\frac{1}{n}\mathbf{H}\mathbf{1}_n\right\|^2 = \|\overline{\mathbf{h}}\|^2.$$

Therefore, we obtain:

$$\|L_2(t)\| \leq \frac{1}{t}\sum_{s=1}^{t}\left\|PR_2^{t-s}P^\top\overline{\mathbf{h}}\right\|.$$

By definition, for any $t \geq s > 0$, $PR_2^{t-s}P^\top\mathbf{1}_n = 0$. Therefore, one has:

$$\|L_2(t)\| \leq \frac{1}{t}\sum_{s=1}^{t}\left\|PR_2^{t-s}P^\top\overline{\mathbf{h}}\right\| \leq \frac{1}{t}\sum_{s=1}^{t}(\lambda_2(2))^{t-s}\|\overline{\mathbf{h}} - \hat{U}_n(H)\mathbf{1}_n\|$$

$$\leq \frac{1}{t}\cdot\frac{1}{1-\lambda_2(2)}\|\overline{\mathbf{h}} - \hat{U}_n(H)\mathbf{1}_n\|,$$

since $\frac{1}{n}\overline{\mathbf{h}}\mathbf{1}_n = \hat{U}_n(H)\mathbf{1}_n$. Similarly, one has:

$$\|L_3(t)\| \leq \frac{1}{t}\cdot\frac{1}{1-\lambda_2(1)}\|\mathbf{H} - \mathbf{1}_n^\top\overline{\mathbf{h}}\|,$$

by definition of $\overline{\mathbf{h}}$ and using $PR_cP^\top\mathbf{1}_{n^2} = 0$. The final term can be upper bounded similarly to previous proofs:

$$\|L_4(t)\| \leq \frac{1}{t}\sum_{s=1}^{t}\left\|PR_2^{t-s}P^\top BP_cR_c^sP_c^\top\mathbf{S}_2(0)\right\|$$

$$\leq \frac{1}{t}\sum_{s=1}^{t}\left\|PR_2^{t-s}P^\top BP_cR_c^sP_c^\top\left(\mathbf{S}_2(0) - \frac{1}{n}\mathbf{1}_n^\top\mathbf{S}_2(0)\right)\right\|$$

$$\leq \frac{1}{t}\left(\sum_{s=1}^{t}(\lambda_2(2))^{t-s}\lambda_2(1)^s\right)\|\mathbf{H} - \mathbf{1}_n^\top\overline{\mathbf{h}}\|.$$

Lemma 1 indicates that $\lambda_2(2) > \lambda_2(1)$, so

$$L_4(t) \leq (\lambda_2(2))^t\|\mathbf{H} - \mathbf{1}_n^\top\overline{\mathbf{h}}\|.$$

Using Lemma 1 and above inequalities, one can finally write:

$$\left\|S_1(t) - \hat{U}_n(H)\mathbf{1}_n\right\| \leq \|L_2(t)\| + \|L_3(t)\| + \|L_4(t)\|$$

$$\leq \frac{c}{t}\|\overline{\mathbf{h}} - \hat{U}_n(H)\mathbf{1}_n\| + \left(\frac{2}{ct} + e^{-ct}\right)\|\mathbf{H} - \mathbf{1}_n^\top\overline{\mathbf{h}}\|,$$

with $c = 1 - \lambda_2(2)$. $\qquad\square$

## C.2 Proof of Theorem 2 (Asynchronous Setting)

For $t > 0$, let us denote as $M(t)$ the expected transition matrix at iteration $t$. With the notation introduced in the synchronous setting, it yields

$$\begin{pmatrix} M_1(t) & M_2(t) \\ 0 & C \end{pmatrix}.$$

The propagation step is unaltered w.r.t. the synchronous case, thus the bottom right block is unmodified. On the other hand, both the transmission step and the averaging step differ: only the selected nodes update their estimators from their associated phantom graph. Therefore, we have:

$$M_2(t) = \frac{1}{|E|} \sum_{(i,j) \in E} \mathbb{E} \begin{pmatrix} \mathbb{I}_{\{1 \in (i,j)\}} \frac{1}{m_1(t)p_1} e_1^\top & 0 & \cdots & & 0 \\ & 0 & \ddots & & \vdots \\ \vdots & & \ddots & & 0 \\ 0 & & \cdots & 0 & \mathbb{I}_{\{n \in (i,j)\}} \frac{1}{m_n(t)p_n} e_n^\top \end{pmatrix}.$$

For any $k \in [n]$ and $t > 0$, $m_k(t)$ is an unbiased estimator of $t$. Moreover, $\sum_{(i,j) \in E} \mathbb{I}_{\{k \in (i,j)\}} = 2d_k$. Therefore, we can write:

$$M_2(t) = \frac{1}{t|E|} \begin{pmatrix} \frac{2d_1}{p_1} e_1^\top & 0 & \cdots & 0 \\ 0 & \ddots & & \vdots \\ \vdots & & \ddots & 0 \\ 0 & \cdots & 0 & \frac{2d_n}{p_n} e_n^\top \end{pmatrix} = \frac{1}{t} \begin{pmatrix} e_1^\top & 0 & \cdots & 0 \\ 0 & \ddots & & \vdots \\ \vdots & & \ddots & 0 \\ 0 & \cdots & 0 & e_n^\top \end{pmatrix} = \frac{B}{t}.$$

Similarly for $M_1(t)$:

$$M_1(t) = W_2(G) - \frac{1}{2t|E|} \sum_{(i,j) \in E} \left( \frac{1}{p_i} e_i(e_i + e_j)^\top + \frac{1}{p_j} e_j(e_i + e_j)^\top \right).$$

Using the definition of $(p_k)_{k \in [n]}$ yields:

$$M_1(t) = W_2(G) - \frac{1}{2t} \left( I_n + D(G)^{-1} A(G) \right).$$

We can now write the expected value of the state vector $S(t)$ similarly to the synchronous setting:

$$\mathbb{E}[\mathbf{S}(t)] = \begin{pmatrix} \mathbf{S}_1(t) \\ \mathbf{S}_2(t) \end{pmatrix} = \begin{pmatrix} \sum_{s=1}^t (M_1(t) \dots M_1(s+1)) \frac{B}{s} C^{s-1} \mathbf{S}_2(0) \\ C^t \mathbf{S}_2(0) \end{pmatrix}.$$

As in the synchronous setting, our proof rely on the eigenvalues of $M(t)$.

*Proof.* For $t > 0$, we have:

$$M_1(t) = W_2(G) - \frac{1}{2t} \left( I_n + D^{-1}(G) A(G) \right) = W_2(G) - \frac{1}{t} I_n + \frac{1}{2t} D(G)^{-1} L(G).$$

Since $M_1(t) \mathbf{1}_n = \left(1 - \frac{1}{t}\right) \mathbf{1}_n$, we have $\|M_1(t)\| \geq 1 - \frac{1}{t}$. Let us denote $\mathrm{Sp}(L(G)) = \{\beta \in \mathbb{R}, \exists \phi \in \mathbb{R}^n, L(G)\phi = \beta\phi\}$. Let $\beta \in \mathrm{Sp}(L(G))$ and $\phi \in \mathbb{R}^n$ a corresponding eigenvector. One can write:

$$M_1(t)\phi = \left( W_2(G) - \frac{1}{t} I_n + \frac{1}{2t} D(G) L(G)^{-1} \right) \phi = \left(1 - \frac{\beta}{|E|} - \frac{1}{t}\right) \phi + \frac{\beta}{2t} D(G)^{-1} \phi$$

$$= \left( \left(1 - \frac{1}{t}\right) I_n - \frac{\beta}{|E|} \left( I_n - \frac{|E| D(G)^{-1}}{2t} \right) \right) \phi.$$

The above matrix is diagonal, therefore we can write:

$$\|M_1(t)\phi\| \leq \max_i \left(1 - \frac{1}{t} - \frac{\beta}{|E|} \left(1 - \frac{|E|}{2d_i t}\right)\right) \|\phi\| = \left(1 - \frac{1}{t} - \frac{\beta}{|E|} \left(1 - \frac{1}{pt}\right)\right) \|\phi\|,$$

where $\bar{p} = \min_i \frac{2d_i}{|E|}$ is the minimum probability of a node being picked at any iteration. Thus, we can see that if $\beta > 0$, $\|M_1(t)\phi\| < (1 - \frac{1}{t})\|\phi\|$ if $t < t_c = \frac{1}{\bar{p}}$. Consequently, if $t \geq t_c$ then $\|\|M_1(t)\|\| = 1 - \frac{1}{t}$. Here, $t_c$ represents the minimum number of iteration needed for every node to have been picked at least once, in expectation.

Let $(\beta_1, \ldots, \beta_n) \in \mathbb{R}$ and $P = (\phi_1, \ldots, \phi_n) \in \mathbb{R}^{n \times n}$ be respectively the eigenvalues and eigenvectors of $L(G)$ (sorted in decreasing order), such that $P$ is the same matrix than the one introduced in Section B. We have:

$$M_1(t)P = PK(t) = P\left(\left(1 - \frac{1}{t}\right)I_n - \frac{1}{|E|}\left(I_n - \frac{|E|}{2t}P^\top D(G)^{-1}P\right)D_{L(G)}\right),$$

where $D_{L(G)} = \mathrm{diag}(\beta_n, \ldots, \beta_1)$. Let $P_1 = (\phi_1, 0, \ldots, 0)$. The matrix $K(t)$ can be rewritten as follows:

$$K(t) = \left(1 - \frac{1}{t}\right)I_n - \frac{1}{|E|}\left(I_n - \frac{|E|}{2t}P^\top D(G)^{-1}P\right)D_{L(G)} = \left(1 - \frac{1}{t}\right)Q + \frac{1}{t}U + R(t),$$

where $Q$, $U$ and $R(t)$ are defined by:

$$\begin{cases} Q &= \mathrm{diag}(1, 0, \ldots, 0), \\ U &= \frac{1}{2}P_1^\top D(G)^{-1}P D_{L(G)}, \\ R(t) &= K(t) - \left(1 - \frac{1}{t}\right)Q - \frac{1}{t}U, \text{ for all } t > 0. \end{cases}$$

Using the fact that $\beta_n = 0$, one can show that $U$ has the form

$$\begin{pmatrix} 0 & * & \cdots & * \\ 0 & & & \\ \vdots & & 0 & \\ 0 & & & \end{pmatrix}.$$

Since $M_1(t)\mathbf{1}_n = \left(1 - \frac{1}{t}\right)\mathbf{1}_n$, we can also show that, for $t > 0$, $R(t)e_1 = \mathbf{0}_n$ and $e_1^\top R(t) = \mathbf{0}_n^\top$.

Let $t > 0$. We can write:

$$M_1(t+1)M_1(t) = PK(t+1)K(t)P^\top$$
$$= P\left(\frac{t}{t+1}Q + \frac{1}{t+1}U + R(t+1)\right)\left(\frac{t-1}{t}Q + \frac{1}{t}U + R(t)\right)P^\top$$
$$= P\left(\frac{t-1}{t+1}Q + \frac{1}{t+1}U\left(I_n + R(t)\right) + R(t+1)R(t)\right)P^\top.$$

Recursively, we obtain, for $t > s > 0$:

$$M_1(t:s) = M_1(t)\ldots M_1(s+1) = P\left(\frac{s}{t}Q + \frac{1}{t}U\sum_{r=s}^{t-1}R(r:s) + R(t:s)\right)P^\top,$$

where we use the convention $R(t-1:t-1) = I_n$.

Let us now write the expected value of the estimates:

$$\mathbb{E}[S_1(t)] = \sum_{s=1}^{t} M_1(t:s)\frac{B}{s}C^{s-1}\mathbf{S}_2(0)$$
$$= \sum_{s=1}^{t} M_1(t:s)\frac{B}{s}P_cQ_cP_c^\top \mathbf{S}_2(0) + \sum_{s=1}^{t} M_1(t:s)\frac{B}{s}P_cR_c^{s-1}P_c^\top \mathbf{S}_2(0)$$
$$= \sum_{s=1}^{t} M_1(t:s)\frac{\overline{\mathbf{h}}}{s} + \sum_{s=1}^{t} M_1(t:s)\frac{B}{s}P_cR_c^{s-1}P_c^\top \mathbf{S}_2(0).$$

The first term can be rewritten as:

$$\sum_{s=1}^{t} M_1(t:s)\frac{\overline{\mathbf{h}}}{s} = \sum_{s=1}^{t} P\left(\frac{s}{t}Q + \frac{U}{t}\sum_{r=s}^{t-1} R(r:s) + R(t:s)\right) P^\top \frac{\overline{\mathbf{h}}}{s}$$

$$= \frac{1}{t}\sum_{s=1}^{t} PQP^\top \overline{\mathbf{h}} + \frac{1}{t}\sum_{s=1}^{t} PU \sum_{r=s}^{t-1} R(r:s)P^\top \frac{\overline{\mathbf{h}}}{s} + \sum_{s=1}^{t} PR(t:s)P^\top \frac{\overline{\mathbf{h}}}{s}$$

$$= \hat{U}_n(H)\mathbf{1}_n + \frac{1}{t}\sum_{s=1}^{t} PU \sum_{r=s}^{t-1} R(r:s)P^\top \frac{\overline{\mathbf{h}}}{s} + \sum_{s=1}^{t} PR(t:s)P^\top \frac{\overline{\mathbf{h}}}{s}$$

$$= \hat{U}_n(H)\mathbf{1}_n + L_1(t) + L_2(t).$$

The second term of the expected estimates can be rewritten as:

$$\sum_{s=1}^{t} M_1(t:s)\frac{B}{s}P_c R_c^{s-1}P_c^\top \mathbf{S}_2(0) = \sum_{s=1}^{t} P\left(\frac{s}{t}Q + \frac{U}{t}\sum_{r=s}^{t-1} R(r:s) + R(t:s)\right) P^\top \frac{B}{s}P_c R_c^{s-1}P_c^\top \mathbf{S}_2(0)$$

$$= L_3(t) + L_4(t) + L_5(t).$$

Now, we need to upper bound $\|L_i(t)\|$ for $1 \leq i \leq 5$. One has:

$$\|L_1(t)\| = \left\|\frac{1}{t}\sum_{s=1}^{t} PU \sum_{r=s}^{t-1} R(r:s)P^\top \frac{\overline{\mathbf{h}}}{s}\right\| = \left\|\frac{1}{t}\sum_{s=1}^{t} PU \sum_{r=s}^{t-1} R(r:s)P^\top \frac{\left(\overline{\mathbf{h}} - \hat{U}_n(H)\mathbf{1}_n\right)}{s}\right\|$$

$$\leq \frac{1}{t}\sum_{s=1}^{t} \left\|PU \sum_{r=s}^{t-1} R(r:s)P^\top \frac{\left(\overline{\mathbf{h}} - \hat{U}_n(H)\mathbf{1}_n\right)}{s}\right\|$$

$$\leq \frac{\|\|U\|\|}{t}\left(\sum_{s=1}^{t} \frac{1}{s}\sum_{r=s}^{t-1} \|\|R(r:s)\|\|\right) \|\overline{\mathbf{h}} - \hat{U}_n(H)\mathbf{1}_n\|.$$

The norm of $U$ can be developed:

$$\|\|U\|\| \leq \frac{1}{2}\|\|D(G)^{-1}\|\| \, \|\|D_{L(G)}\|\| = \frac{\beta_1}{|E|\overline{p}}.$$

Moreover, for $2 \leq i \leq n$, one has:

$$\|R(t)e_i\| = \left\|\left(1 - \frac{1}{t}\right)e_i - \frac{\beta_{n-i+1}}{|E|}e_i + \frac{\beta_{n-i+1}}{2t}P_{2:}^\top D(G)^{-1}\phi_i\right\|$$

$$\leq \left(\left(1 - \frac{1}{t}\right) - \frac{\beta_{n-i+1}}{|E|}\right)\|e_i\| + \left\|\frac{\beta_{n-i+1}}{2t}P_{2:}^\top D(G)^{-1}\phi_i\right\|$$

$$\leq \left(\left(1 - \frac{1}{t}\right) - \frac{\beta_{n-i+1}}{|E|}\left(1 - \frac{1}{\overline{p}t}\right)\right)\|e_i\|.$$

For $t > 0$, let us define $\mu_R(t)$ by:

$$\mu_R(t) = \left(1 - \frac{1}{t}\right) - \frac{\beta_{n-1}}{|E|}\left(1 - \frac{1}{\overline{p}t}\right).$$

We then have, for any $t > 0$, $\|\|R(t)\|\| < \mu_R(t)$. Thus,

$$\|L_1(t)\| \leq \frac{\beta_1}{|E|\overline{p}t}\left(\sum_{s=1}^{t} \frac{1}{s}\sum_{r=s}^{t-1} \mu_R(r:s)\right) \|\overline{\mathbf{h}} - \hat{U}_n(H)\mathbf{1}_n\|.$$

Also,

$$\|L_2(t)\| = \left\|\sum_{s=1}^{t} PR(t:s)P^\top \frac{\overline{\mathbf{h}} - \hat{U}_n(H)\mathbf{1}_n}{s}\right\| \leq \sum_{s=1}^{t} \frac{\mu_R(t:s)}{s}\|\overline{\mathbf{h}} - \hat{U}_n(H)\mathbf{1}_n\|$$

A reasoning similar to the synchronous setting can be applied to $L_3(t)$:

$$\|L_3(t)\| = \frac{1}{t}\left\|\sum_{s=1}^{t} PQP^\top \frac{B}{s} P_c R_c^{s-1} P_c^\top \mathbf{h}\right\| \leq \frac{1}{t} \cdot \frac{1}{1-\lambda_2(1)} \|\mathbf{H} - \bar{\mathbf{h}}\mathbf{1}_n^\top\|.$$

Concerning $L_4(t)$, one can write:

$$\|L_4(t)\| = \frac{1}{t}\left\|\sum_{s=1}^{t}\left(U\sum_{r=s}^{t-1} R(r:s)\right)\frac{B}{s}P_c R_c^{s-1}P_c^\top \mathbf{S}_2(0)\right\|$$

$$\leq \frac{\beta_1}{|E|\bar{p}t}\sum_{s=1}^{t}\frac{1}{s}\left(\sum_{r=s}^{t-1}\mu_R(r:s)\right)(\lambda_2(1))^{s-1}\|\mathbf{H} - \bar{\mathbf{h}}\mathbf{1}_n^\top\|.$$

Similarly, one has:

$$\|L_5(t)\| \leq \|\mathbf{H} - \bar{\mathbf{h}}\mathbf{1}_n^\top\|\sum_{s=1}^{t}\frac{\mu_R(t:s)}{s}(\lambda_2(1))^{s-1}.$$

Now, for $t > s > 1$, one only need to find appropriate rates on $\sum_{s=1}^{t}\frac{1}{s}\mu_R(t:s)$ and $\sum_{s=1}^{t}\frac{1}{s}\sum_{r=s}^{t-1}\mu_R(r:s)$ to conclude. Here, for $t > 1$, $\mu_R(t)$ can be rewritten as follow:

$$\mu_R(t) = \left(\frac{t-1}{t}\right)\lambda_2(1)\left(1 + (1-\lambda_2(1))\frac{c}{t}\right),$$

with $c = \frac{1}{\lambda_2(1)\bar{p}} - 1$. If $c < 1$, one cas use a reasoning similar to the synchronous setting and conclude. However, $c$ is often greater than 1. In this case, one has:

$$\mu_R(t) \leq \left(\frac{t-1}{t}\right)\lambda_2(1)\left(1 + \frac{c}{t}\right).$$

For $t > s > 0$, the product $\mu_R(t:s)$ can then be bounded as follow:

$$\mu_R(t:s) \leq \frac{s}{t}\lambda_2(1)^{t-s}\left(1 + \frac{c}{t-1}\right)\ldots\left(1 + \frac{c}{s}\right).$$

Using the definition of $t_c$, it is clear that, for $t \geq t_c$, one has $\lambda_2(1)(1 + \frac{c}{t-1}) < 1$. We can use this result to upper bound $\sum_{s=1}^{t}\frac{\mu_R(t:s)}{s}$ with a geometric series:

$$\sum_{s=1}^{t}\frac{1}{s}\mu_R(t:s) \leq \frac{1}{t}\sum_{s=1}^{t}\lambda_2(1)^{t-s}\left(1 + \frac{c}{t-1}\right)\ldots\left(1 + \frac{c}{s}\right)$$

$$\leq \frac{1}{t}\sum_{s=t_c+1}^{t}\lambda_2(1)^{t-s}\left(1 + \frac{c}{t_c}\right)^{t-s} + \frac{1}{t}\sum_{s=1}^{t_c}\lambda_2(1)^{t-s}\left(1 + \frac{c}{t-1}\right)\ldots\left(1 + \frac{c}{s}\right)$$

$$\leq \frac{1}{t} \cdot \frac{1}{1-\mu_c} + \frac{t_c}{t}(1+c)^{t_c}e^{-(1-\lambda_c)(t-t_c)},$$

where $\mu_c = \lambda_2(1)\left(1 + \frac{c}{t_c}\right)$. Therefore, we have that $\sum_{s=1}^{t}\frac{1}{s}\mu_R(t:s) = O(1/t)$. Let us now focus on the second bound. For $t > t_c$ and $1 < s < t$, one has:

$$\sum_{s=1}^{t}\frac{1}{s}\sum_{r=s}^{t-1}\mu_R(r:s) \leq \sum_{s=1}^{t_c}\frac{1}{s}\sum_{r=s}^{t_c}\mu_R(r:s) + \sum_{s=1}^{t_c}\frac{1}{s}\sum_{r=t_c+1}^{t-1}\mu_R(r:s) + \sum_{s=t_c+1}^{t}\frac{1}{s}\sum_{r=s}^{t-1}\mu_R(r:s)$$

$$\leq \sum_{s=1}^{t_c}\sum_{r=s}^{t_c}\frac{1}{r}\lambda_2(2)^{r-s}(1+c)^{r-s} + \sum_{s=1}^{t_c}\sum_{r=t_c+1}^{t-1}\frac{\mu_c^{r-s}}{r} + \sum_{s=t_c+1}^{t}\sum_{r=s}^{t-1}\frac{\mu_c^{r-s}}{r}$$

$$\leq t_c\sum_{r=1}^{t_c}\frac{1}{r}\lambda_2(2)^r(1+c)^r + t_c\mu_c^{-t_c}\sum_{r=t_c+1}^{t-1}\frac{\mu_c^r}{r} + \sum_{s=1}^{t}\frac{1}{s}\sum_{r=0}^{s-1}\mu_c^r$$

$$\leq t_c(1+c)^{t_c} - t_c\mu_c^{-t_c}\log(1-\mu_c) + \frac{1}{1-\mu_c}\sum_{s=1}^{t}\frac{1}{s}$$

$$\leq t_c(1+c)^{t_c} + \frac{t_c\mu_c^{t-c}}{1-\mu_c} + \frac{1}{1-\mu_c}\log(t+1).$$

**Algorithm 4** U2-gossip [18]

**Require:** Each node $k$ holds observation $X_k$
1: Each node initializes $Y_k^{(1)} \leftarrow X_k, Y_k^{(2)} \leftarrow X_k, Z_k \leftarrow 0$
2: **for** $t = 1, 2, \ldots$ **do**
3:      **for** $p = 1, \ldots, n$ **do**
4:          $Z_p \leftarrow \frac{t-1}{t} Z_p + \frac{1}{t} H(Y_p^{(1)}, Y_p^{(2)})$
5:      **end for**
6:      Draw $(i, j)$ uniformly at random from $E$
7:      Nodes $i$ and $j$ swap their first auxiliary observations: $Y_i^{(1)} \leftrightarrow Y_j^{(1)}$
8:      Draw $(k, l)$ uniformly at random from $E$
9:      Nodes $k$ and $l$ swap their second auxiliary observations: $Y_k^{(2)} \leftrightarrow Y_l^{(2)}$
10: **end for**

Thus, $\sum_{s=1}^{t} \frac{1}{s} \sum_{r=s}^{t-1} \mu_R(r:s) = O(\log t)$.

Using these results and the previous expressions of $L_1(t), \ldots, L_5(t)$, one can conclude that, for $t > 1$, $\|\mathbb{E}[Z(t)] - \hat{U}_n(H) \mathbf{1}_n\| = O(\log t / t)$.      $\square$

## D    U2-gossip Algorithm

U2-gossip [18] is an alternative approach for computing $U$-statistics. In this algorithm, each node stores two auxiliary observations that are propagated using independent random walks. These two auxiliary observations will be used for estimating the $U$-statistic – see Algorithm 4 for details. This algorithm has an $O(1/t)$ convergence rate, as stated in Theorem 4.

Let $k \in [n]$. At iteration $t = 1$, the auxiliary observations have not been swapped yet, so the expected estimator $\mathbb{E}[Z_k]$ is simply updated as follow:

$$\mathbb{E}[Z_k(1)] = \mathbb{E}[Z_k(0)] + e_k^\top \mathbf{H} e_k.$$

Then, at the end of the iteration, auxiliary observations are randomly swapped. Therefore, one has:

$$\mathbb{E}[Z_k(2)] = \frac{1}{2} \mathbb{E}[Z_k(1)] + \frac{1}{2} \left( W_1(G) e_k^\top \right) \mathbf{H} W_1(G) e_k.$$

Using recursion, we can write, for any $t > 0$ and any $k \in [n]$:

$$\mathbb{E}[Z_k(t)] = \sum_{s=0}^{t-1} e_k^\top W_1(G)^s \mathbf{H} W_1(G)^s e_k. \tag{19}$$

We can now state a convergence result for Algorithm 4.

**Theorem 4.** *Let us assume that $G$ is connected and non bipartite. Then, for $\mathbf{Z}(t)$ defined in Algorithm 4, we have that for all $k \in [n]$:*

$$\lim_{t \to +\infty} \mathbb{E}[Z_k(t)] = \frac{1}{n^2} \sum_{1 \le i,j \le n} H(X_i, X_j) = \hat{U}_n(H)$$

*Moreover, for any $t > 0$,*

$$\left\| \mathbb{E}[\mathbf{Z}(t)] - \hat{U}_n(H) \mathbf{1}_n \right\| \le \frac{\sqrt{n}}{t} \left( \frac{2}{1 - \lambda_2(1)} \| \overline{\mathbf{h}} - \hat{U}_n(H) \mathbf{1}_n \| + \frac{1}{1 - \lambda_2(1)^2} \| \mathbf{H} - \overline{\mathbf{h}} \mathbf{1}_n^\top \| \right),$$

*where $\lambda_2(1)$ is the second largest eigenvalue of $W_1(G)$.*

*Proof.* Let $k \in [n]$ and $t > 0$. Using the expression of $\mathbb{E}[Z_k(t)]$ established in (19), one has:

$$\mathbb{E}[Z_k(t)] = \frac{1}{t} \sum_{s=0}^{t} e_k^\top W_1(G)^s \mathbf{H} W_1(G)^s e_k = \frac{1}{t} \sum_{s=0}^{t} e_k^\top P^\top D_1^s P \mathbf{H} P D_1^s P^\top e_k,$$

where $P$ is the eigenvectors matrix introduced in Section B and $D_1 = \mathrm{diag}(\lambda_1(1), \ldots, \lambda_n(1))$. Similarly to previous proofs, we split $D_1 = Q_1 + P_1$ where $Q_1 = \mathrm{diag}(1, 0, \ldots, 0)$ and $R_1 = \mathrm{diag}(0, \lambda_2(1), \ldots, \lambda_n(1))$. Now, we can write $\mathbb{E}[Z_k(t)] = L_1(t) + L_2(t) + L_3(t) + L_4(t)$ with $L_1(t), L_2(t), L_3(t)$ and $L_4(t)$ defined as follows:

$$
\left\{
\begin{array}{rcl}
L_1(t) & = & \frac{1}{t} \sum_{s=1}^{t} e_k^\top P^\top Q_1^s P \mathbf{H} P Q_1^s P^\top e_k \\
L_2(t) & = & \frac{1}{t} \sum_{s=1}^{t} e_k^\top P^\top R_1^s P \mathbf{H} P Q_1^s P^\top e_k \\
L_3(t) & = & \frac{1}{t} \sum_{s=1}^{t} e_k^\top P^\top Q_1^s P \mathbf{H} P R_1^s P^\top e_k \\
L_4(t) & = & \frac{1}{t} \sum_{s=1}^{t} e_k^\top P^\top R_1^s P \mathbf{H} P R_1^s P^\top e_k
\end{array}
\right. .
$$

The first term can be rewritten:

$$
L_1(t) = e_k^\top P^\top Q_1 P \mathbf{H} P Q_1 P^\top e_k = \frac{1}{n^2} \mathbf{1}_n^\top \mathbf{H} \mathbf{1}_n = \hat{U}_n(H).
$$

Then, one has:

$$
\begin{aligned}
|L_2(t)| &\leq \frac{1}{t} \sum_{s=0}^{t} \| e_k^\top P R_1^s P^\top \mathbf{H} P Q_1 P^\top e_k \| \leq \frac{1}{t} \sum_{s=0}^{t} \| P R_1^s P^\top \overline{\mathbf{h}} \| \\
&\leq \frac{1}{t} \sum_{s=0}^{t} (\lambda_2(1))^s \| \overline{\mathbf{h}} - \hat{U}_n(H) \mathbf{1}_n \| \leq \frac{1}{t} \cdot \frac{1}{1 - \lambda_2(1)} \| \overline{\mathbf{h}} - \hat{U}_n(H) \mathbf{1}_n \|,
\end{aligned}
$$

since $\lambda_2(1) < 1$. Similarly, we have $|L_3(t)| \leq \frac{1}{t} \cdot \frac{\lambda_2(1)}{1 - \lambda_2(1)} \| \overline{\mathbf{h}} - \hat{U}_n(H) \mathbf{1}_n \|$. The final term $L_4(t)$ can be bounded as follow:

$$
\begin{aligned}
L_4(t) &\leq \frac{1}{t} \sum_{s=0}^{t} \left| e_k^\top P R_1^s P^\top \mathbf{H} P Q_1 P^\top e_k \right| = \frac{1}{t} \sum_{s=0}^{t} \left| e_k^\top P R_1^s P^\top \left( \mathbf{H} - \mathbf{1}_n \overline{\mathbf{h}}^\top \right) P Q_1 P^\top e_k \right| \\
&\leq \frac{1}{t} \sum_{s=0}^{t} (\lambda_2(1))^{2s} \left\| \mathbf{H} - \mathbf{1}_n \overline{\mathbf{h}}^\top \right\| \leq \frac{1}{t} \cdot \frac{1}{1 - (\lambda_2(1))^2} \left\| \mathbf{H} - \mathbf{1}_n \overline{\mathbf{h}}^\top \right\|.
\end{aligned}
$$

With above relations, the expected difference can be bounded as follow:

$$
\begin{aligned}
\left| \mathbb{E}[Z_k(t)] - \hat{U}_n(H) \right| &\leq |L_2(t)| + |L_3(t)| + |L_4(t)| \\
&\leq \frac{1}{t} \cdot \frac{2}{1 - \lambda_2(1)} \left\| \overline{\mathbf{h}} - \hat{U}_n(H) \mathbf{1}_n \right\| + \frac{1}{t} \cdot \frac{1}{1 - (\lambda_2(1))^2} \left\| \mathbf{H} - \mathbf{1}_n \overline{\mathbf{h}}^\top \right\|.
\end{aligned}
$$

Finally, we can conclude:

$$
\begin{aligned}
\left\| \mathbb{E}[\mathbf{Z}(t)] - \hat{U}_n(H) \right\| &\leq \sqrt{n} \max_{k \in [n]} \left| \mathbb{E}[Z_k(t)] - \hat{U}_n(H) \right| \\
&\leq \frac{\sqrt{n}}{t} \cdot \frac{2}{1 - \lambda_2(1)} \left\| \overline{\mathbf{h}} - \hat{U}_n(H) \mathbf{1}_n \right\| + \frac{\sqrt{n}}{t} \cdot \frac{1}{1 - (\lambda_2(1))^2} \left\| \mathbf{H} - \mathbf{1}_n \overline{\mathbf{h}}^\top \right\|.
\end{aligned}
$$

$\square$

# E   Comparison to Baseline Methods

In this section, we use the within-cluster point scatter problem studied in Section 5 to compare our algorithms to two — more naive — baseline methods, described below.

**Gossip-flooding baseline.**   This baseline uses the same communication scheme than GoSta-async (Algorithm 2) to flood observations across the network, but we assume that each node has enough memory to store all the observations it receives. At each iteration, each selected node picks a random observation among those it currently holds and send it to the other (tagged with the node which initially possessed it, to avoid storing duplicates). The local estimates are computed using the subset of observations available at each node (the averaging step is removed).

Figure 4: Comparison to the gossip-flooding baseline.

Figure 5: Comparison to the master-node baseline. One unit of data corresponds to one observation coordinate.

Figure 4 shows the evolution over time of the average relative error and the associated standard deviation *across nodes* for this baseline and GoSta-async on the networks introduced in Section 5. On average, GoSta-async slightly outperforms Gossip-flooding, and this difference gets larger as the network connectivity decreases. The variance of the estimates across nodes is also lower for GoSta-async. This confirms the interest of averaging the estimates, and shows that assuming large memory at each node is not necessary to achieve good performance. Finally, note that updating the local estimate of a node is computationally much cheaper in GoSta-async (only one function evaluation) than in Gossip-flooding (as many function evaluations as there are observations on the node).

**Master-node baseline.** This baseline has access to a master node $\mathcal{M}$ which is connected to every other node in the network. Initially, at $t = 0$, each node $i \in [n]$ sends its observation $X_i$ to $\mathcal{M}$. Then, at each iteration $t \in [n]$, $\mathcal{M}$ sends observation $X_t$ to every node of the network. As in Gossip-flooding, the estimates are computed using the subset of observations available at each node. The performance of this baseline does not depend on the original network, since communication goes through the master-node $\mathcal{M}$. This allows us to compare our approach to the ideal scenario of a "star" network, where a central node can efficiently broadcast information to the entire network.

For a fair comparison with GoSta-async, we evaluate the methods with respect to the communication cost instead of the number of iterations. Figure 5 shows the evolution of the average relative error for this baseline and GoSta-async. We can see that the Master-node baseline performs better early on, but GoSta-async quickly catches up (the better the connectivity, the sooner). This shows that our data propagation and averaging mechanisms compensate well for the lack of central node.