[Reviews · NeurIPS 2015]

Submitted by Assigned_Reviewer_1

This paper presents new Gossip algorithms for computation of U-statistics. The proposed algorithms have faster convergence rate compared to state-of-the-art (U2-Gossip algorithm from Pelckmans, Suykens'09) as well as work in the more realistic asynchronous settings. The experimental results further demonstrate the superiority of the proposed algorithms compared to existing algorithms.

The paper is well written, the problem is well motivated and the authors have well positioned their work w.r.t the existing literature. As far I can see, the proposed algorithms are novel, and superior in terms of convergence and cost (memory and communication in comparison to the U2-gossip). Also, the proposed algorithm works for asynchronous settings.

Apart from theoretical guarantees and analysis, the authors have also done extensive empirical evaluations over two datasets and compared to the baselines. This further strengthens the contributions of the paper.
Summary: This paper presents new Gossip Algorithms for computing U-statistics, with faster convergence rate as well as for the realistic asynchronous settings. Furthermore, the algorithms are evaluated and compared to state-of-art via extensive experiments.

Submitted by Assigned_Reviewer_2

This paper is about computing higher-order statistics measured over pairs of observations, where the observations themselves are distributed across a network of nodes. These U-statistics include many quantities of interest such as the sample variance, correlation and scatter coefficients, and other aggregate measures. The main goal of the paper is to propose a gossip-style decentralized aggregation/averaging algorithm to compute these statistics.

Overall, this is a nice contribution to the literature on gossip algorithms, but the authors should make a better connection to existing approaches and ideas.

This is a "heavy" review. I did not have time to verify the correctness of the material in the supplement.

Comments

1) The main trick on the algorithm side is that when two nodes exchange their estimate values and average them, they also swap a secondary state variable representing their initial observation. This means that the initial observations are each performing a random walk (albeit coupled). Mentioning this more clearly in the main manuscript would help and also would motivate the analysis method via the multiple copies of the graph.

2) There has been a lot of work on gossip approaches for computing statistics beyond the references cited by the authors. A snapshot of the state of the art from a few years ago is the Dimakis et al. Proc. IEEE paper. Situating the current work in the context of the works mentioned there might be more helpful than the somewhat cursory explanation in the current manuscript.

3) On page 4 it might be worth clarifying what the induced distribution on edges is from the "node wakes up, contacts neighbor" process. The phrase "drawing a random edge of the network" does not specify the distribution.

4) There is a natural other approach to this problem, which is to allow more memory at each node and essentially use the same contact process to have each node flood its own value across the network. Nodes store the received observations (tagged with the node who initially possessed it). At each time nodes can calculate the U statistic on the subset of the data that they possess. How would this compare to the current method? This may help show why the averaging can be helpful, or show the loss in performance when the memory at each node is limited. Without this comparison it's not clear that the rate 1/t is good or bad. For expanders the flooding method may be quite good.

5) The authors should take a look at the work on lifting constructions for accelerating gossip algorithms, as it seems quite related to their construction in the paper. See for example:

K. Jung, D. Shah, and J. Shin, "Fast gossip through lifted Markov chains," in Proc. Allerton Conf. on Comm., Control, and Comp., Urbana-Champaign, IL, Sep. 2007. W. Li and H. Dai, "Location-aided fast distributed consensus," in IEEE Transactions on Information Theory 56(12), 2010. F. Chen, L. Lovasz, and I. Pak, "Lifting markov chains to speed up mixing," in Proceedings of the thirty-first annual ACM symposium on Theory of computing. ACM, 1999, pp. 275-281.

6) Analysis of gossip algorithms tend to make a worst-case assumption on the initial data distribution. That is, the guarantee is over all initial data vectors $X$. This is not clear in the problem statement of the statement of the theorem. What is the assumption on the data distribution and what are the guarantees on the error as a function of the initial data?

Small things - p1: The bipartite definition using (S \times T) \cup (T \times S) implies directed graphs. - p4: "let us stress out that" --> "we stress that" - p4: "awakes and contact" -- > "awakes and contacts" - p6: "requires to initiate" --> "requires initiating" - p7: "allows to create" --> "allows us to create"

AFTER AUTHOR FEEDBACK: I think the clarifications and modifications suggested by the authors are good. I am revising my initial score of 7 up to an 8.

I think that they should still take a close look at the lifting constructions (esp. LADA) and modify the related work.

Furthermore, the experiments should include the baseline comparisons.

Summary: This is a nicely presented, approach to extending gossip methods to compute more complex statistics in distributed settings such as sensor networks. The mathematical formulation and connection to prior work is a little unclear. Assuming the authors make a nice revision and address the comments raise in this review, I recommend accepting this paper.

Submitted by Assigned_Reviewer_3

The paper is well motivated and written. The results of their approach is presented on 2 datasets (evaluated on avg relative AUC error). Compared to the baseline (U2-gossip), their method performs much better in varied scenarios (graphs). The paper gives convergence bounds for their method. - I would like to see see run-times per iteration for the proposed and U2-gossip methods. If each iteration of the proposed method is much slower than U2-gossip, then the paper would be of limited utility.
Summary: The paper proposes sync. and async. randomized gossip algorithms for computing U-statistics.

Submitted by Assigned_Reviewer_4

This paper talks about the problem of computation of average statistics in a distributed environment. The algorithm itself is quite intuitive and may be lacking clear novelty - though the proofs are comprehensive.

One weakness of the paper is the it does not compare with any baseline measures. I am sure several intuitive baselines can be generated based on randomized algorithms and some should be considered for experimentation.
Summary: The paper solves an interesting problem but the evaluation is weak and there are no baseline measures used.

Submitted by Assigned_Reviewer_5

The paper presents a synchronous and an asynchronous gossip algorithm for U-statistics. The paper derives the techniques and shows that they achieve convergence rates of 1/t and log t / t, respectively.

The paper is well organized and shows results that demonstrate the performance. The area it studies is of some interest and the presented technique is thus of use to different researchers.
Summary: This paper introduces new gossip algorithms for the estimation of U-statistics that has favorable convergence rates. Experiments show its performance advantages over other approaches and illustrate its accuracy.

Author Feedback
Author rebuttal: We thank the reviewers for their careful reading and helpful comments. We will take their remarks into account to improve the quality and clarity of the manuscript. Below, we first address the two main concerns raised, then provide specific answers to each reviewer.

** Comparison to baselines **

As noted by Reviewer2 and Reviewer8, it would be interesting to compare our approach to simple baselines that assume the nodes have enough memory to store many observations (which may not be the case in practice). We are currently running experiments with the baseline suggested by Reviewer2. Moreover, we have already performed a comparison to another baseline which is a bit easier to implement and interesting as well. It relies on a "master node" which is connected to all other nodes and can thus be used to send an observation to every node for a relatively small communication cost. At each iteration, a node is drawn (uniformly) at random and its observation is sent to every node through the master node, and each node computes an estimate using the U-statistic computed on the subsample of observations it currently holds. Of course, in practice a master node is rarely available, but we use it to compare our approach to an ideal "centralized" case. We compared the estimation quality of this baseline to that of our approach w.r.t. their communication cost. We observed that our approach achieves similar performance for well-connected graphs (Watts-Strogatz and complete graph), and for the 2D-grid the degradation of performance is minor. We will add the comparisons to these baselines to the paper.

** Scale of experiments **

Reviewer5 and Reviewer9 argue that the scale of experiments is too small. Convergence on larger networks typically requires more iterations (as shown in Figure 3a for the complete graph), which is partly because larger networks tend to have smaller spectral gap. For this reason, we focused our efforts on varying the graph connectivity rather than its size, and we do not expect that increasing the size of the network will affect our conclusions as to the relative performance of GoSta and U2-gossip.
Also note that our algorithm is able to run asynchronously, therefore in a given time frame, more iterations are performed in a large graph than in a small one, compensating for the larger number of iterations needed.
Then, the number of nodes we considered is of the same order w.r.t papers with similar theoretical flavor in the field: for instance in Duchi et al. 2012, the number of nodes considered grows up to n=1000.
That being said, we have already run experiments with n=10000 nodes and obtained conclusions similar to the one we have already presented. We will add them to the paper.

** Reviewer2 **

We thank the reviewer for his/her thorough review and interesting references. As suggested we will clarify and improve the related work section. Regarding comment (6), we do not make any assumption on the data distribution, hence our results hold in the worst-case (i.e., for any initial vectors). Note however that there is some data dependence in Theorem 1 through the norm terms. These terms are small (meaning faster convergence) if all the values to be averaged in Eq. (1) are concentrated around the average value U_n(H).

** Reviewer4 **

In general, providing a runtime comparison of decentralized algorithms is difficult as it heavily depends on whether the limiting factor is the computational power of the nodes or the network communication infrastructure for the particular problem of interest. That being said, the local computation in GoSta and U2-Gossip is very simple (averaging scalars) and can be ignored. U2-Gossip however requires twice as much communication as GoSta and opens two different communication channels at each iteration, so the runtime per iteration of our approach is significantly smaller when communication is the bottleneck.

** Reviewer9 **

Comment (a): Initially the nodes cannot compute even a single term of the U-statistic, so some amount of data propagation is unavoidable. Our contribution is to provide a way to estimate the U-statistic only using pairwise and asynchronous exchange of observations, which is generally practical even for large networks of simple units. In practice, thanks to the averaging step, the nodes converge to a good estimate of the desired quantity after seeing only a small fraction of the total dataset.
Comment (b): We presented our algorithm with one observation by node to get a simpler analysis. The extension sketched in the paper for the case where each node has several observations could certainly be improved. For instance, several observations could be exchanged at each iteration (depending on the communication constraints) to avoid the problem raised by the reviewer. Given the current length of the paper, we believe that a proper analysis of this extension is best left for future work.